# Semantic Similarity Based Label Augmentation for Visual Classification

**Yu Cao**
eBay Group Services GmbH
14532 Kleinmachnow, Germany
ycao8@ebay.com

## Abstract

Real-world applications may present visual categories for which examples are many but a definition is elusive. When data augmentation helps little and hand-crafted heuristics fail to warrant weak supervision, similarity remains a simple but effective guide for augmenting training labels. This paper showcases, for recognizing clutters in e-commerce photography, how similarity between visual representations pre-trained to capture semantics provides quality supervision. Our approach significantly improves on the label propagation baseline, with classification precision and recall above 0.8 in most cases.

## 1 Introduction

Language-vision models with contrastive pre-training like CLIP (Radford et al., 2021) allow us to measure text-image similarity and thereby *semantic* similarity among images themselves—think of it as mediated via latent prompts. This similarity can prove useful in label augmentation for visual classification. As a case study, we consider clutter recognition in e-commerce photography, where we have only a handful of labeled photos and a myriad of unlabeled ones.

Clutters can be *anything* adding no values but distracting customers from viewing the product; see Figure 1 for examples. The concept, compared to usual object categories like dogs or cars, is vague enough to defy common image augmentation techniques, which are unlikely to deliver a diversity of examples naturally seen in reality.

Nor do we have working heuristics for weak supervision (WS). A texture descriptor like standard deviation (Gonzalez, 2018) may somewhat predict $-$CLUTTER, but its precision with $+$CLUTTER is below random guess, as it ignores higher-level semantics.

## 2 Related work

**Data augmentation** For visual classification, data augmentation synthesizes new images out of existing labeled examples and retains their labels. Shorten & Khoshgoftaar (2019) offer a thorough survey on rule-based and learned augmentation techniques for deep learning models.

**Label augmentation** This technique underlies WS (Ratner et al., 2016; 2017): it generates noisy labels for unlabeled data by exploiting heuristics and sometimes a few ground truths. Besides reporting recent progress in WS, Solmaz et al. (2022) propose to extend the coverage of labeling functions using data similarities. Label propagation (Zhu & Ghahramani, 2002) is a closely related technique that draws on similarity to label unlabeled data.

## 3 Approach

We perform neighbor search with Faiss (Johnson et al., 2019) to iteratively grow the classes $\mathbb{P}$ and $\mathbb{N}$ of $\pm$CLUTTER images. After embedding the universe of all labeled and unlabeled images and holding out a labeled subset $\mathbb{H}$, we set $\mathbb{P}_0$ as $+$CLUTTER seeds and $\mathbb{N}_0$ as $-$CLUTTER seeds. Let

$n_k(\boldsymbol{x})$ be the set of $k$ nearest neighbors of $\boldsymbol{x}$ other than $\boldsymbol{x}$ in the universe; for $i \geq 1$,

$$\mathbb{P}'_i = \bigcup_{\boldsymbol{x} \in \mathbb{P}_{i-1}} n_k(\boldsymbol{x}) \setminus \bigcup_{0 \leq j < i} \mathbb{P}_j \setminus \bigcup_{0 \leq j < i} \mathbb{N}_j,$$

$$\mathbb{N}'_i = \bigcup_{\boldsymbol{x} \in \mathbb{N}_{i-1}} n_k(\boldsymbol{x}) \setminus \bigcup_{0 \leq j < i} \mathbb{P}_j \setminus \bigcup_{0 \leq j < i} \mathbb{N}_j,$$

$$\mathbb{P}_i = \mathbb{P}'_i \setminus \mathbb{N}'_i \setminus \mathbb{H},$$

$$\mathbb{N}_i = \mathbb{N}'_i \setminus \mathbb{P}'_i \setminus \mathbb{H}.$$

Iteration stops as soon as $|\mathbb{P}_{i-1}| < |\mathbb{P}_i|$ or $|\mathbb{N}_{i-1}| < |\mathbb{N}_i|$, since the search henceforth will return increasingly more points that have appeared before. The union of all steps $\mathbb{P}_0, ..., \mathbb{P}_i$ gives the augmented class $\mathbb{P}$ of +CLUTTER images. $\mathbb{N}$ is similarly obtained.

Here the hyperparameter $k$ and the iteration stopping criterion give us some control over the similarity between seeds and sampled neighbors; taking the neighbors of neighbors into account contributes to the diversity; throughout the process we have kept $\mathbb{P}$ and $\mathbb{N}$ disjoint.

## 4 EXPERIMENTS

We validated the quality of similarity-augmented labels with three learning experiments, resp. for clutter recognition in the *Home & Garden* (e.g. sofas), *Fashion* (e.g. sneakers), and *Electronics* (e.g. laptops) departments.

**Data**  Of each department 10K images were labeled by 3 up to 7 workers after due instruction on clutters; a total of 125,284 judgments were collected with an agreement rate above 0.8 more than 70% of time. A mean opinion score (MOS) $\in [0,1]$ arises from summing the responses *Yes* = 1, *No* = 0, *Unsure* = 0.5 with the workers' credibility as weights. A MOS $\geq 0.5$ reads +CLUTTER; otherwise −CLUTTER if all workers agreed the image showed exactly one product.

We take 30% of labeled examples for validation and augment the rest with $k = 10$. To compare, an unlabeled set of equal size is sampled for label propagation[1]: in the adjacency matrix $\boldsymbol{A}_{u,v} = 1$ *iff* $v$ is one of the 10 nearest neighbors of $u$; the steps of propagation track those of augmentation. Table 1 gives the scale of our datasets.

**Classification**  We added on top of frozen CLIP and, for comparison, DINO[2] (Caron et al., 2021) a single dense layer and trained it with and without augmented or propagated labels.

As Table 2 shows, CLIP-based augmentation achieves the best result; it by far outperforms propagation and, less so, the DINO baselines. Augmentation tops propagation regardless of the embedding, but its benefit over training without extra labels manifests only with CLIP, even if by a thin edge (which was expected, as either way validation loss converged quickly). This suggests that DINO may induce a similarity different in nature from the semantic similarity we ascribe to CLIP.

## 5 CONCLUSION

We have shown that semantic similarity can be directly used to generate quality labels. While our method is generally applicable to novel visual categories, future work is needed to explore how larger-scale augmentation—comparable to the scale of typical datasets on which modern computer vision networks are trained—may benefit the generalization of models deeper than ours.

### URM STATEMENT

The author acknowledges that they meet the URM criteria of ICLR 2023 Tiny Papers Track.

---

[1]Implemented at `https://github.com/dmlc/dgl`.
[2]Implemented at `https://github.com/huggingface/transformers`.

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

## A  APPENDIX

Table 1: The scale of the $\pm$CLUTTER seed, augmented, and validation sets.

| | Home & Garden | | Fashion | | Electronics | |
|---|---|---|---|---|---|---|
| | +CLUTTER | −CLUTTER | +CLUTTER | −CLUTTER | +CLUTTER | −CLUTTER |
| Seeds | 1.3K | 4.2K | 1.2K | 4.8K | 2.8K | 3.3K |
| Augmented$_{\text{CLIP}}$ | 19.6K | 54.3K | 50.4K | 170.8K | 70.0K | 76.5K |
| Augmented$_{\text{DINO}}$ | 17.7K | 58.6K | 38.8K | 152.8K | 68.4K | 78.3K |
| Validation | .5K | 1.8K | .5K | 2.1K | 1.2K | 1.4K |

Table 2: Validation per ±(CLUTTER) P(recision), R(ecall), and the H(armonic) M(ean) of all metrics, with color-coded column max/min.

| | Home & Garden | | | | Fashion | | | | Electronics | | | | |
|---|---|---|---|---|---|---|---|---|---|---|---|---|---|
| | +P | +R | -P | -R | +P | +R | -P | -R | +P | +R | -P | -R | HM |
| CLIP/seeds | .85 | .64 | .90 | .97 | .83 | .58 | .90 | .97 | .81 | .85 | .87 | .84 | .82 |
| CLIP/aug | .80 | .68 | .91 | .95 | .82 | .60 | .91 | .97 | .79 | .90 | .90 | .80 | .82 |
| CLIP/prop | .83 | .48 | .86 | .97 | .90 | .38 | .86 | .99 | .79 | .60 | .72 | .87 | .71 |
| DINO/seeds | .82 | .57 | .88 | .96 | .83 | .58 | .90 | .97 | .78 | .89 | .90 | .79 | .80 |
| DINO/aug | .79 | .46 | .85 | .96 | .80 | .54 | .89 | .97 | .79 | .83 | .85 | .81 | .76 |
| DINO/prop | .89 | .23 | .81 | .99 | .87 | .28 | .84 | .99 | .82 | .54 | .70 | .90 | .59 |

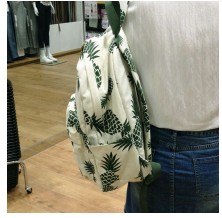

+CLUTTER

*Clutter due to mirror reflection and human model.*

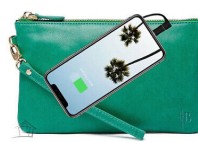

+CLUTTER

*The prop iPhone blocks and isn't sold with the item (purse).*

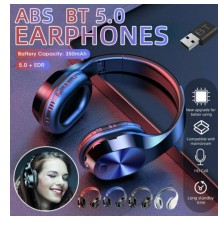

+CLUTTER

*Clutter due to artificial texts and artworks.*

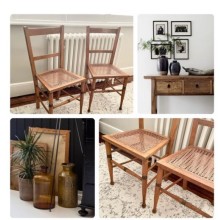

+CLUTTER

*Clutter due to photo collage.*

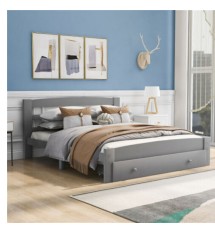

−CLUTTER

*Staging the item (bed frame) in a cozy bedroom adds values.*

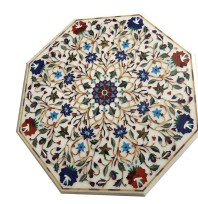

−CLUTTER

*The complex pattern belongs to the item.*

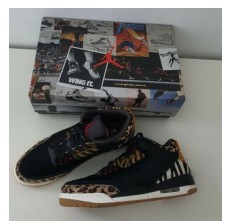

−CLUTTER

*The complex pattern belongs to the item's container.*

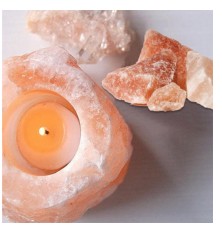

−CLUTTER

*Plain background.*

Figure 1: Examples of cluttered (+CLUTTER) and non-cluttered (−CLUTTER) e-commerce images.

