# OpenReview forum: "Semantic Similarity Based Label Augmentation for Visual Classification"
_ICLR.cc/2023/TinyPapers — Submitted to Tiny Papers @ ICLR 2023_

### Official Review · Reviewer_Z386 · 2023-03-26

**Confidence:** 5

**Summary Of Contributions:**

This paper proposed a techniques for clutter recognization based on the similarity of CLIP models.

**Rating:**

Great Start (GS): a submission which meets some of the reviewing criteria but has room for improvement

**Strengths And Weaknesses:**

S1: This paper tackles one interesting challenge where the definition of a positive class can be elusive, such as clutter, where there can be much different clutter for the e-commerce scenario.

S2: The performance of the developed method seems to be good.

W1: I would say the paper does not fully utilize the CLIP model, as the full potential of CLIP is the ability for matching between language and vision, the argument in the paper is that language and vision matching also provides a good similarity measure for image only, I would suggest to also add a self-supervised baseline like DINO to further enhance this point.

W2: Some other similarity-based techniques should be reported, such as label propagation, how about first do label propagation on the dataset and then training the model with the propagated labels?


**Suggested Changes:**

C1: Replacing the CLIP model with a self-supervised model like DINO can help to see the potential of the proposed method.

C2: I would strongly suggest the author implement label propagation, and compare the proposed similarity-based method with them.

C3: Another baseline comparison I would recommend is to add a comparison with CLIP using text prompts like "a cluttered photo of {CLS}".

C4: The baseline in table 1 is to use salient object detection, I would suggest to add some discussion of why this is a good baseline.

---

### Author Response · Authors · 2023-05-30
**For archival**

 I wish to opt-in for archival

---

### Meta-Review · Area_Chair_wEnK · 2023-04-06

**Recommendation:** Invite to present
**Confidence:** 5

**Metareview:**

This paper received one review, the reviewer thinks that the paper tackles one interesting problem and the developed method seems to perform good.
Some extension experiments are suggested by the reviewer, which is recommended for the author to include in the final version.
Although the reviewer is concerned about the lack of more experiments, the paper itself is clear and reproducible.


**Summary:**

A method for recognizing clutters in images is proposed, the method is presented clearly and has shown the effectiveness, more experiments are suggested by the reviewer.

**Reason For Not Giving A Higher Recommendation:**

It seems that the lack of a more comprehensive comparison limits the potential of this work.


**Reason For Not Giving A Lower Recommendation:**

The paper itself is clear and reproducible.

---

### Decision · Program_Chairs · 2023-04-10

Invite to present